
# Integrating macroseismic intensity distributions by a probabilistic approach: an application in Italy

Andrea Antonucci[14], Andrea Rovida[1], Vera D'Amico[2] and Dario Albarello[3]

[1]Istituto Nazionale di Geofisica e Vulcanologia, Milano, 20133, Italy
[2]Istituto Nazionale di Geofisica e Vulcanologia, Pisa, 56125, Italy
[3]Department of Physics, Earth and Environmental Sciences, University of Siena, Siena, 53100, Italy
[4]Department of Earth Sciences, University of Pisa, Pisa, 56126, Italy

*Correspondence to*: Andrea Antonucci (andrea.antonucci@ingv.it)

**Abstract.** The geographic distribution of earthquake effects quantified in terms of macroseismic intensities, the so-called macroseismic field, provides basic information for several scopes including source characterization of pre-instrumental earthquakes and risk analysis. Macroseismic fields of past earthquakes as inferred from historical documentation may present spatial gaps, due to the incompleteness of the available information. We present a probabilistic approach aimed at integrating incomplete intensity distributions by considering the Bayesian combination of estimates provided by Intensity Prediction Equations (IPEs) and data documented at nearby localities, accounting for the relevant uncertainties and the discrete and ordinal nature of intensity data. The performance of the proposed methodology is tested at 28 Italian localities with long and rich seismic histories, and for two well-known strong earthquakes (i.e., 1980 Southern Italy and 2009 Central Italy events). A possible application of the approach is also illustrated relative to a sixteenth century earthquake in Northern Apennines.

## 1 Introduction

Characterizing earthquake effects on the anthropic environment is of paramount importance for estimating seismic risks and planning prevention politics. This characterization is performed by classifying earthquake effects according to macroseismic scales. Each macroseismic scale considers a set of scenarios, twelve in the most used scales in Europe (i.e., MCS, Sieberg, 1932; MSK, Medvedev et al., 1964; EMS98, Grünthal, 1998), ordered in terms of increasing severity of the effects. Through macroseismic scales observed seismic effects concerning human behavior, damage to buildings and geomorphological phenomena at a site are compared with the scenarios proposed in the scale to assess an intensity value. An intensity value, referred to a specific earthquake and a specific place, identified through its geographic coordinates defines the Intensity Data Point (IDP in the following). The spatial distribution of IDPs is considered for the characterization of earthquake sources (i.e., estimates of epicentral location and magnitude) in the absence of instrumental data (e.g., Bakun and Wentworth, 1997; Gasperini et al., 1999; 2010; Provost and Scotti, 2020). Collecting these parameters in homogeneous seismic catalogues (e.g., Fäh et al., 2011; Stucchi et al., 2013; Manchuel et al., 2017; Rovida et al., 2019; 2020) is a key element to provide a seismic characterization of a region, and this information represents a basic tool for seismic hazard estimates (e.g., Stucchi et al., 2011;



Woessner et al., 2015; Meletti et al., 2021). In particular, in countries with rich macroseismic data (e.g., Italy and France) the histories of documented earthquake effects at a site can be consistently used for local seismic hazard assessment (e.g., Albarello and Mucciarelli, 2002; D'Amico and Albarello, 2008). Moreover, macroseismic intensity can be useful to check the outcomes of probabilistic seismic hazard assessments (Stirling and Petersen, 2006; Mucciarelli et al., 2008; Rey et al., 2018), especially in countries where the historical record is much longer than the instrumental one.

Retrieving seismological data from documentary information requires specific methodologies and expertise (e.g., Guidoboni and Stucchi, 1993) and presents several criticalities mainly due to the incompleteness of the documentation (e.g., Albarello et al., 2001; Swiss Seismological Service, 2002; Stucchi et al., 2004; Hough and Martin, 2021). The probability to retrieve such documentation depends on the period, size and location of the event and is hampered by the survival of sources and the capability of retrieving and analyzing them (e.g., Albini and Rovida, 2018; Albini, 2020a; 2020b). This implies that intensity distributions of historical events may present important gaps, which depend also on the density and importance of the settlements affected by the earthquakes.

To fill these gaps, documented seismic effects may be integrated with "synthetic" intensities, which can be estimated in different ways. Until the second half of the twentieth century, qualitative contouring procedures were used to draw isoseismal maps (e.g., Shebalin, 1974; Postpischl, 1980; Barbano et al., 1980; Ferrari and Postpischl, 1985; Patané and Imposa, 1985), in which hand-drawn isoseismals bounded areas enclosing sites with intensity overcoming any given threshold (Musson and Cecić, 2012). This form of regularization aims at reconstructing a general radiation pattern for historical earthquakes but is affected by biases induced by the conceptual background of the "tracer". To overcome this drawback, some authors (Ambraseys and Douglas, 2004; Rey et al., 2018) proposed geostatistical approaches (e.g., Olea, 1999) to identify areas affected by similar seismic effects. They applied the kriging spatial interpolation technique to compute the expected values of macroseismic intensity through a semivariogram that describes the correlation between neighboring IDPs. This kind of approach, however, disregards the inherent ordinal and discrete nature of intensity data, which requires specific formalizations to account for uncertainty affecting intensity estimates (see, e.g., Magri et al., 1994; Albarello and Mucciarelli, 2002).

An alternative approach to obtain synthetic intensities makes use of Intensity Prediction Equations (IPEs), which provide the possible intensity values at any site as a function of epicentral distance and maximum or epicentral intensity or magnitude (e.g., Pasolini et al., 2008; Sørensen et al., 2009; Allen et al., 2012; Rotondi et al., 2016). The limitation of this approach is the hypothesis that the radiation pattern of seismic waves from the source is the only responsible for the intensity at a site, disregarding lateral heterogeneities induced by the fracture process and geological/geomorphological features.

To account for these features we present an alternative probabilistic approach, which improves the one proposed by Albarello et al. (2007) and D'Amico and Albarello (2008). The key element is a combination, through a Bayesian approach, of probabilistic estimates provided by an IPE constrained by observed intensities that are spatially close to the site of interest.


The proposed procedure is described at first, then it is tested on a set of localities and macroseismic fields included in the Italian Macroseismic Database DBMI15 (Locati et al., 2019).

**2 Methodology**

In the frame of a coherent Bayesian formalization, the proposed procedure combines intensities estimated at a site with an IPE with observed intensities at neighboring localities for the same earthquake, taking into account the inherent uncertainty. To this purpose, considering any *l-th* event, the discrete probability density distribution $p_l(I_s/I_v)$ is computed, to associate to each possible intensity value $I_s$ at the site $s$ a probability value conditioned by the occurrence of effects of intensity $I_v$ at any other

site $v$:

$$p_l(I_s|I_v) = p_l(I_s) \frac{q(I_v|I_s)}{\sum_{I=1}^{12} p_l(I) q(I_v|I)} \tag{1}$$

Here $p_l(I_s)$ represents the "prior" probability density which is deduced from an IPE by using the epicentral parameters (location, epicentral intensity or magnitude, etc.) of the *l-th* event. In general, the most common IPEs (in their probabilistic formulation) have the form

$$S(I_s|I_e, D) = prob[I \geq I_s|I_e, D] = \frac{1}{\sigma\sqrt{2\pi}} \int_{I_s-0.5}^{\infty} e^{-\frac{(J-\mu(I_e,D))^2}{\sigma^2}} dJ \tag{2}$$

(Albarello and D'Amico, 2004) where the average $\mu$ is a function of the epicentral distance $D$ and the intensity at the epicenter $I_e$ (or, eventually, the estimated magnitude). Both average $\mu$ and standard deviation $\sigma$ are determined from the statistical analysis of the available information (e.g., Pasolini et al., 2008 for Italy). To account for the uncertainty affecting epicentral intensity, the marginal probability can be computed as

$$P_l(I_s) = \sum_{I_e=1}^{I_{max}} g(I_e) S(I_s|I_e, D) \tag{3}$$

where $g$ is the probability distribution which expresses the uncertainty affecting the epicentral intensity and Imax is the upper bound of the adopted macroseismic scale (e.g., 12 for the MCS scale). The probability density $p_l(I_s)$ can be computed in the form

$$\begin{cases} p_l(I_s < I_{max}) = P_l(I_s) - P_l(I_s + 1) \\ \quad p_l(I_s = I_{max}) = P_l(I_{max}) \end{cases} \tag{4}$$

It is worth noting that Eq. (1) can be iteratively applied when an increasing number of neighboring sites is considered to constrain intensity. This can be simply performed by substituting the "prior" distribution with the output of the preceding estimate. The key element of Eq. (1) is the conditional probability density $q(I_v/I_s)$, which expresses the correlation between intensity values at neighboring localities. In other terms, such a probability density represents the "smoothness" of the macroseismic field and plays the role of covariance in classical geostatistics. More specifically, $q(I_v/I_s)$ expresses the



constraining power of $I_s$ on $I_v$. According to Albarello et al. (2007), $q(I_v/I_s)$ can be estimated empirically by considering observed intensity distributions. To this purpose, data provided by the Italian Macroseismic Database DBMI15 (Locati et al., 2019) were considered as a case study.

## 3 Assessing the spatial variability of macroseismic data in Italy

### 3.1 The Italian macroseismic database DBMI15

The long tradition of historical macroseismic investigation in Italy has produced a wealth of studies and data on the seismic history of the country and neighboring areas. All such studies are collected and organized in the Italian Archive of Historical Earthquake Data – ASMI (https://emidius.mi.ingv.it/ASMI/index_en.htm, Rovida et al., 2017), which grants access to the information on more than 6200 earthquakes occurred in the Italian area from 461 B.C. to 2019. The data gathered in ASMI are of several typologies and formats, and provide a large number of intensity data from different sources, such as macroseismic

bulletins, online databases, and many scientific papers and reports. The different information collected in ASMI for each earthquake requires a careful comparison in order to identify the reference study among those available for the compilation of the Italian Macroseismic Database DBMI. The current release of the latter (DBMI15, https://emidius.mi.ingv.it/CPTI15-DBMI15/, here considered in its version 2.0, Locati et al., 2019) is the result of the specific selection of these data according to the content and quality of each study and to the number and spatial distribution of intensity data. DBMI15 makes available

123756 IDPs related to 3219 Italian earthquakes in the time-window 1000-2017, and referred to 20000 localities, of which 15332 are in Italy. DBMI15 results from 189 different studies, and the intensity data they provide are not homogenous as regards the geographic coordinates and the standards used for assessing macroseismic intensities. For this purpose, a series of operations were performed in order to obtain an homogenous set of intensity data: i) a unique gazetteer, covering the whole national territory was adopted in order to match the position of a locality with the macroseismic observation, and ii) a standard

to express the macroseismic intensity (e.g., 6, 6-7, 7 MCS) was defined and non-conventional descriptive codes (e.g., "D" for damage, or "F" for felt) were adopted when the available information is not sufficient for assessing an intensity value. DBMI15 allows direct access to seismic histories of Italian localities and provides data upon which the macroseismic parameters of the Parametric Catalogue of Italian Earthquakes - CPTI15 (https://emidius.mi.ingv.it/CPTI15-DBMI15/query_eq/, Rovida et al., 2019; 2020) are built.

Although the guidelines of the European Macroseismic Scale EMS98 (Grünthal, 1998) recommend the users to preserve the integer character of the intensity scale and avoid forms such as "6.5" or "6½" or "6+", in many studies intensity data are listed as intermediate values in order to express uncertainty affecting the intensity estimate. This is also the solution adopted in DBMI15. Following D'Amico and Albarello (2008), intensity data can be classified in two categories: "certain data" (one single intensity value $I'$, e.g., 6) and "uncertain data" (pair of values $I'$- $I''$, e.g., 6-7). In case intensity $I_v$ is uncertain between

two contiguous values $I_v'$ and $I_v''$ (e.g., $I_v$ = 6-7), Eq. (1) becomes:


$$p_l(I_s|I_v) = 0.5\, p_l(I_s|I'_v) + 0.5\, p_l(I_s|I''_v) \tag{5}$$

where an equal probability is assigned to the hypotheses $I_v = I'$ and $I_v = I''$. This is the way "uncertain" intensity values contained in DBMI15 are treated in the following analysis.

### 3.2 Results

To estimate the probability $q(I_v/I_s)$ in Eq. (1) one can consider the relative frequencies of the differences between intensity values at pair of sites affected by the same event. Such probability is expected to monotonically decrease with the distance between the sites, and above any distance threshold $q(I_v/I_s) \approx q(I_v)$, i.e., $I_s$ becomes not informative about $I_v$. The closer the sites considered are, the higher the informative power of $I_s$ on $I_v$ is expected to be, because closer sites possibly share also the same seismostratigraphical and geomorphological conditions. On the other hand, since the selected sites correspond to urbanized

areas (settlements, villages, towns, etc.), each conventionally represented with the geographical coordinates of a single point, there is a lower limit to the distances between considered sites, which depends on the size and density of urbanized areas. Moreover, distances in the estimate of $q(I_v/I_s)$ should be large enough to include at least two sites.

In order to evaluate the optimal distance threshold to characterize $q(I_v/I_s)$, the geographic distribution of the 15332 Italian localities in DBMI15 has been investigated. In particular, for each locality, the number of localities within a set of possible

distance thresholds has been computed. Figure 1 shows that there are significant parts of the Italian territory (mainly in southern areas) where the mutual distance of localities is larger than 10 km. On the other hand, for all the sites (except for 9 localities on small islands), there is at least another locality within 20 km. The latter was thus selected as the reference distance threshold for the characterization of $q(I_v/I_s)$.




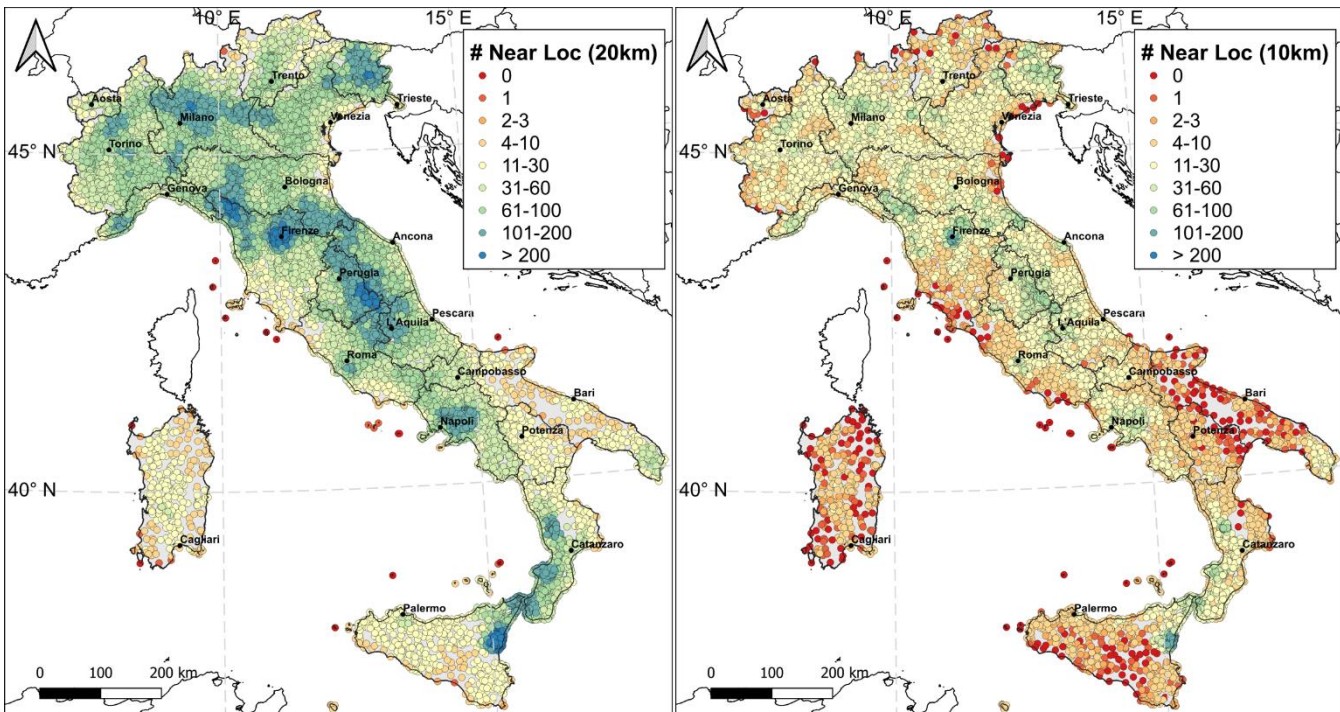

**Figure 1: Number of neighboring localities within 10 km (left) and 20 km (right) for all the Italian localities in DBMI15.**

To this purpose, a dataset derived from DBMI15 has been defined by selecting 546 earthquakes with at least 10 IDPs with intensity greater than or equal to 5 MCS. These earthquakes occurred in the period 1117-2017 CE and are well distributed over the whole Italian territory. From the intensity distributions of these earthquakes, we discarded:

i)      non-numerical macroseismic observations (e.g., "Damage" or "Felt");

ii)     data related to unidentified localities or large areas (see Locati et al. (2019) for details);

iii)    macroseismic observations related to earthquakes with epicenter inside the active volcanic areas (i.e., Mt. Etna and Campanian volcanoes), due to the faster attenuation observed in these zones with respect to the rest of Italy (e.g., Carletti and Gasperini, 2003).

We obtained 58062 IDPs with intensity ranging from 1-2 to 11 MCS (Fig.2), referred to more than 12500 Italian localities.


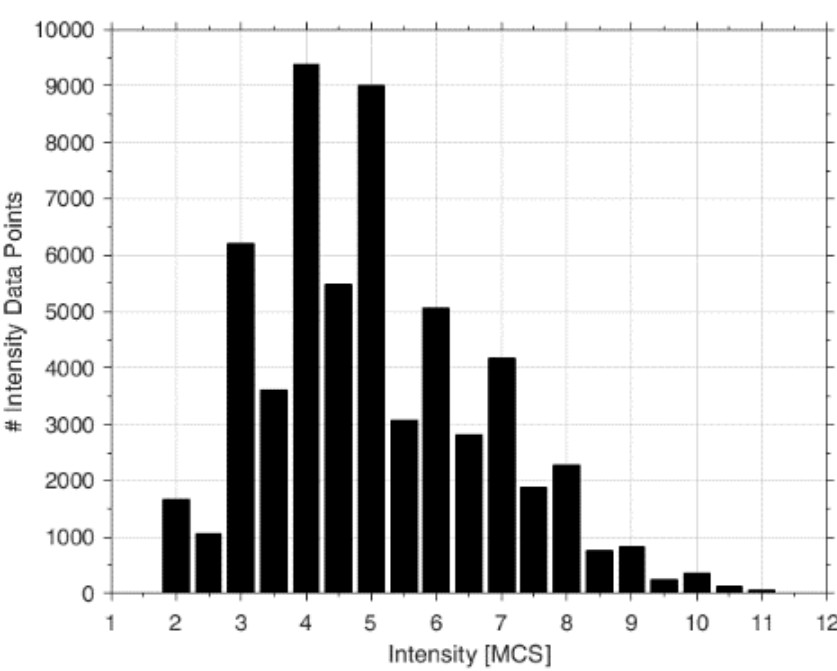


**Figure 2: Frequencies of selected intensity values related to the 546 earthquakes used in the analysis.**

The conditional probability $q(I_v/I_s)$ in Eq. (1) was estimated from the relative frequencies of the differences between Iv and Is computed for each of the 546 selected earthquakes. In this analysis $I_v$ and $I_s$ represent any pair of intensity values observed at neighboring localities (i.e., within a distance of 20 km). If the intensity values $I_s$ and $I_v$ are both "uncertain" (e.g., 6-7), the two

adjacent integer degrees (i.e., 6 and 7) are considered as equiprobable and the differences between the four intensity values are computed; if both Is and Iv are "certain" values (e.g., 7), the difference is counted four times (Albarello et al., 2007). In general, one has

$$q(I_v|I_s) = q(\Delta I|I_v,I_s) \tag{6}$$

where $\Delta I = I_v - I_s$ and the dependence of both $I_v$ and $I_s$ is due to the lack of a defined metrics for intensity degrees. As a

preliminary step, we assume that

$$q(\Delta I|I_v,I_s) = q(\Delta I) \tag{7}$$

which corresponds to the assumption of a linear metrics for intensity values. This hypothesis will be tested in the following.

Two different analyses were performed in order to estimate the frequency distribution $q(\Delta I)$ from the residuals $(I_v - I_s)$ considering: i) only the nearest IDP within 20 km, and ii) all the IDPs within 20 km. Table 1 reports the values of $q(\Delta I)$

expressed as the relative frequency of cases where, for each earthquake, site intensity $I_s$ differs by $\Delta I$ from the intensity $I_v$ observed at the nearest locality $(q(\Delta I)_{near})$ and at all localities $(q(\Delta I)_{all})$ within 20 km. The evident differences between the two analyses is that for $q(\Delta I)_{all}$ the probability distribution results less peaked at $\Delta I = 0$ and thus broader than $q(\Delta I)_{near}$. Figure 3





shows the effects of releasing the assumption in Eq. (7). To evaluate this aspect, we tested the dependence of $q(\Delta I)$ on $I_s$. The results show that the probability of having the same intensity in the nearest locality is slightly higher for $I_s$ equal to 7 and 8 while this probability tends to decrease for $I_s$ equal to 6, 4, 9 and 5, respectively. The outcomes of this analysis seem to be almost independent from the intensity $I_s$. As a consequence, we consider the approximation in Eq. (7) reliable.

In both analyses, we then verified the impact on the results of the distance among localities. Figure 4 shows the effect of distance on $q(\Delta I)$. This effect is quite weak and only concerns the probability of observing the same intensity at the two sites ($\Delta I = 0$). Considering only the nearest locality within 20 km (Fig. 4a), the frequency of $\Delta I = 0$ decreases from around 52% for the distance range 0-5 km to 42% for the range 15-20 km. When all localities within 20 km are considered (Fig. 4b), this relative frequency decreases from 48% to 36% for the range 0-5 km and 15-20 km, respectively.

**Table 1: Values of $q(I_v|I_s)$ expressed as the probability that site intensity $I_s$ differs by $\Delta I$ from the intensity $I_v$ observed at the nearest locality within 20 km ($q(\Delta I)_{near}$) and at all the neighboring localities within 20 km ($q(\Delta I)_{all}$).**

| $\Delta I = (I_v - I_s)$ | $q(\Delta I)_{near}$ | $q(\Delta I)_{all}$ |
|---|---|---|
| - 6 | 0.00002 | 0.00002 |
| - 5 | 0.00005 | 0.00030 |
| - 4 | 0.00050 | 0.00199 |
| - 3 | 0.00516 | 0.01335 |
| - 2 | 0.03736 | 0.06250 |
| - 1 | 0.20128 | 0.22177 |
| 0 | 0.49367 | 0.40016 |
| 1 | 0.21105 | 0.22177 |
| 2 | 0.04353 | 0.06250 |
| 3 | 0.00664 | 0.01335 |
| 4 | 0.00068 | 0.00199 |
| 5 | 0.00007 | 0.00030 |
| 6 | 0.00000 | 0.00002 |


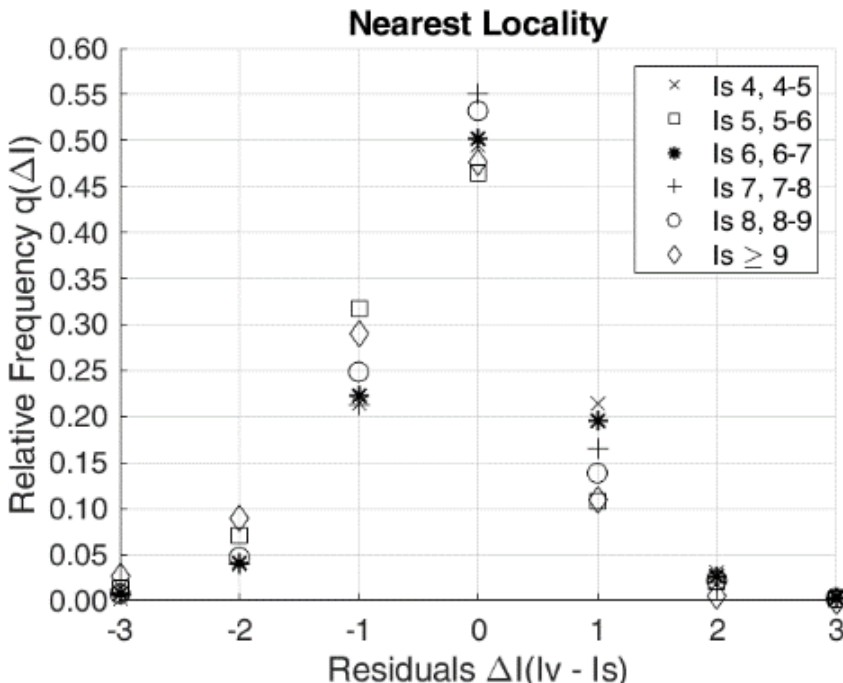

**Figure 3: Relative frequency of *q(ΔI)* as a function of intensity *I_s* for the nearest locality within 20 km.**

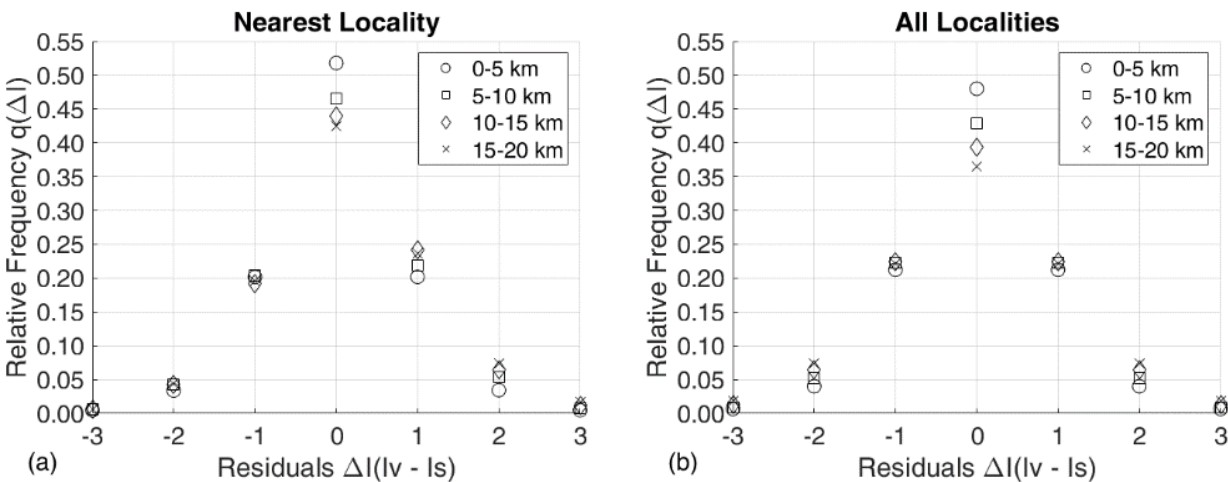

**Figure 4: Relative frequency of *q(ΔI)* as a function of different distance ranges for (a) the nearest locality within 20 km, and (b) all the localities within 20 km.**


## 4 Testing

### 4.1 Testing procedure

To test the effectiveness of this procedure, the probability distributions in Eq. (1) were computed for a set of Italian localities and then compared with available observations. The estimated probabilities $p_l(I_s/I_v)$, derived at each *j-th* site for each *l-th* earthquake, were used to calculate the predicted number of occurrences for each intensity degree $I_s$ (*Npred*) over the total of the *M* sites and *N* earthquakes considered:

$$Npred = \sum_{j=1}^{M} \sum_{l=1}^{N} p_{lj}(I_s|I_v) \tag{8}$$

The predicted values (*Npred*) can be compared with the observed number of occurrences (*Nobs*) for the same intensity degree $I_s$. If the observed intensity value is "certain" (e.g., 6), it can be expressed as:

$$Nobs = \sum_{j=1}^{M} \sum_{l=1}^{N} I_{slj} \tag{9}$$

In case the $I_s$ value is "uncertain" (e.g., 6-7), an equal probability (0.5) was assigned to the two adjacent integer degrees. The Central Limit Theorem was used to check the consistency between predicted and observed values. The statistical test *Z* follows

the standardized Gauss distribution:

$$Z = \frac{(Nobs - Npred)}{\sigma pred} \tag{10}$$

The standard deviation (*σpred*) associated to the predicted values was estimated as described in Albarello and D'Amico (2005):

$$\sigma pred = \sqrt{\sum_{j=1}^{M} \sum_{l=1}^{N} \{p_{lj}(I_s|I_v)[1 - p_{lj}(I_s|I_v)]\}} \tag{11}$$

Equation (10) can be used to evaluate the statistical significance of the discrepancy between predicted (*Npred*) and observed

(*Nobs*) values. According to Albarello and D'Amico (2005), when |Z| is greater than 2, the resulting discrepancy can be considered statistically significant at the 5% confidence level.

### 4.2 Application

The above approach was applied to 28 localities with at least 40 intensity data in DBMI15 homogeneously distributed over the Italian territory. For each locality and for each earthquake, we computed the probability $p_l(I_s/I_v)$ with Eq. (1) using the

probability distribution $q(\Delta I)$ in Eq. (6) derived from all near localities within 20 km (Table 1), by excluding the intensity observed at the site of concern. We then estimated the predicted number of occurrences for each intensity degree $I_s$ for all sites and earthquakes through Eq. (8) and compared it with the observed occurrences (Eq. 10).

The probability $p_l(I_s/I_v)$ has been computed with three different analyses for the prior distribution $p_l(I_s)$ and for $q(\Delta I)$:





- a) using a uniform distribution over the intensity range 2-11 for $p_l(I_s)$ and the intensity observed at the nearest locality within 20 km (i.e., probability $q(\Delta I)_{near}$ in Table 1);

- b) using a uniform distribution over the intensity range 2-11 for $p_l(I_s)$ and iteratively considering in Eq. (1) the intensities observed at all the localities within 20 km (i.e., probability $q(\Delta I)_{all}$ in Table 1);

- c) using as $p_l(I_s)$ the probability computed through an IPE and the intensities observed at all the localities within 20 km (i.e., probability $q(\Delta I)_{all}$ in Table 1); the IPE defined for Italy by Pasolini et al. (2008) and recalibrated by Lolli et al. (2019) with IDPs from DBMI15 and earthquake parameters provided by CPTI15 (Rovida et al., 2019; 2020) was used.

Using Eq. (10) the number of observed occurrences (*Nobs*) for a given intensity value $I_s$ was compared with the predicted number (*Npred*) derived for the three possible choices (*a, b, c*) of the prior distribution $p_l(I_s)$ and $q(\Delta I)$ for the 28 selected localities (Table 2). The differences in percentage between predicted and observed values were computed, and expressed as [(1- *Npred/Nobs*) * 100]. *Z* represents the standardized Gaussian statistics: if $|Z| > 2$, the resulting discrepancy can be considered statistically significant (probability 0.05).

**Table 2: Observed (*Nobs*) and predicted (*Npred*) number of intensity values for each analysis (*a*, *b*, *c*) with their differences in percentage (Diff), and the results of the Z test (Z).**

| $I_s$ | Nobs | Npred a | Npred b | Npred c | Diff (%) a | Diff (%) b | Diff (%) c | Z a | Z b | Z c |
|---|---|---|---|---|---|---|---|---|---|---|
| *1* | 0 | 0.00 | 0.00 | 13.10 | \ | \ | \ | \ | \ | \ |
| *2* | 58.0 | 135.06 | 84.17 | 80.98 | -132.85 | -45.12 | -39.62 | -8.43 | -3.57 | -3.34 |
| *3* | 252.0 | 245.08 | 226.69 | 263.03 | 2.74 | 10.04 | -4.38 | 0.55 | 2.31 | -1.01 |
| *4* | 352.5 | 295.63 | 377.01 | 406.33 | 16.13 | -6.95 | -15.27 | 4.09 | -1.94 | -4.47 |
| *5* | 337.0 | 268.02 | 275.68 | 257.00 | 20.47 | 18.20 | 23.74 | 5.20 | 5.40 | 8.01 |
| *6* | 151.0 | 174.79 | 160.79 | 140.30 | -15.76 | -6.48 | 7.08 | -2.14 | -1.10 | 1.47 |
| *7* | 96.0 | 104.53 | 108.26 | 90.29 | -8.88 | -12.77 | 5.95 | -1.00 | -1.86 | 1.10 |
| *8* | 42.0 | 52.68 | 51.39 | 42.61 | -25.44 | -22.35 | -1.45 | -1.72 | -2.08 | -0.19 |
| *9* | 17.0 | 22.66 | 19.26 | 13.17 | -33.31 | -13.28 | 22.53 | -1.38 | -0.78 | 1.99 |
| *≥ 10* | 5.5 | 12.55 | 7.75 | 4.19 | -128.12 | -40.91 | 23.82 | -1.69 | -0.89 | 1.01 |



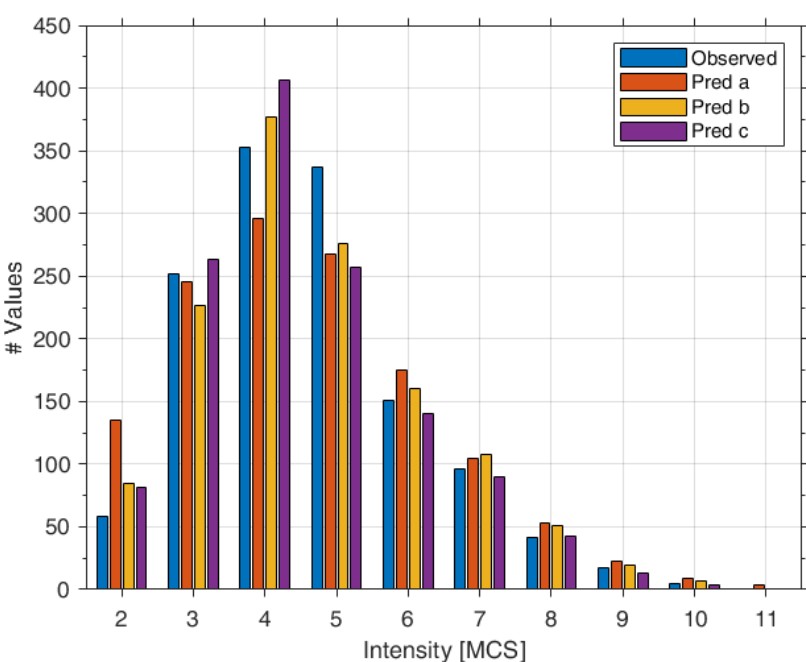

**Figure 5: Observed (blue bar) and predicted number of intensity values for analysis *a* (orange bar), *b* (yellow bar) and *c* (violet bar).**

Table 2 shows that, for analysis *c*, the differences between observed and predicted values are less than 10% for intensity 3, 6, 7, 8. For analysis *a* and *b*, this is verified only for intensity 3, 7 and 4, 6 respectively. Results for intensity 2 and 11 cannot be considered significant due to the strong data incompleteness for the former (Fig. 5) and to the lack of data (only 1 observation available) for the latter. Furthermore, for intensity 5 there is a significant underestimation of the observed intensities in all the analyses, whereas for intensity 4 analysis *b* and *c* tend to overestimate the observed values (see Fig. 5 and *Z* values in Tab. 2). These outcomes indicate that the number of predicted values (*Npred*) is consistent with the number of observed occurrences (*Nobs*) at the 28 test localities. Among the three analyses, analysis *c* is more effective than the others, although the discrepancies expressed with the *Z* test are statistically significant for intensity 4 and 5. However, this may depend on the selected dataset, because DBMI15 contains only earthquakes with maximum intensity greater than or equal to 5 (Locati et al., 2019).

To verify the impact of using this procedure rather than the IPE alone to predict intensity values, a comparison test was carried out for two well-documented recent Italian earthquakes, i.e., the Mw 6.3 event occurred on 6 April 2009 in the L'Aquila area (Central Italy) and the Mw 6.8 Irpinia (Southern Italy) earthquake of 23 November 1980. For both earthquakes we computed the differences between the observed intensity values as reported in DBMI15 (Fig. 6 and Fig. 7, respectively) and: i) the intensity values computed with the IPE by Pasolini et al. (2008) recalibrated by Lolli et al. (2019) and ii) the intensity values estimated with the proposed procedure following the analysis *c* described above. In both cases, the modal value of each probability distribution computed by Equation [1] for all the sites was considered.



For the 2009 L'Aquila earthquake, Figure 6a shows that for 33 out of 315 sites (11%) the values predicted with the IPE alone are equal to the observed ones, and for 206 sites (65%) the predicted intensities differ by more than 1 intensity degree from the observations. The results obtained with our procedure (Fig. 6b) show a higher predictive performance because 218 sites

255    (69%) present the same predicted intensity as the observed value and, for 288 sites (91%), the differences are within 1 intensity degree. For the 1980 Irpinia earthquake, Figure 7 shows that the intensity values predicted with the IPE alone are equal to the observed ones for 652 out of the 1202 considered sites (54%), whereas using the proposed methodology these sites become 822 (68%). A difference of 1 intensity degree between the predicted values and the observed ones is shown at 478 sites (40%) with the IPE alone (Fig. 7a), whereas at 350 sites (29%) with our procedure (Fig. 7b).

260    This test demonstrates that the intensity values obtained by means of the proposed procedure better reproduce the observed intensities than using the IPE alone. In fact, more than the 90% of differences between predicted and observed intensity values are within 1 intensity degree, that is the uncertainty associated to any macroseismic intensity assessment.

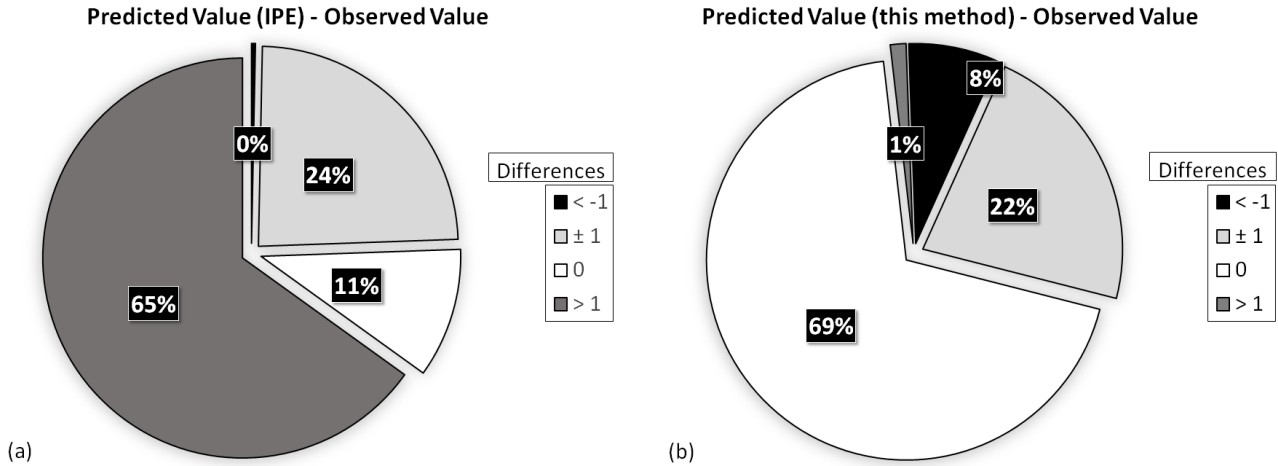

**Figure 6: Differences between the observed intensity values, as reported in DBMI15 (Galli and Camassi, 2009), for the 2009 L'Aquila**
265    **earthquake and the intensity values computed with (a) the IPE alone (Pasolini et al., 2008; Lolli et al., 2019) and (b) the proposed procedure.**





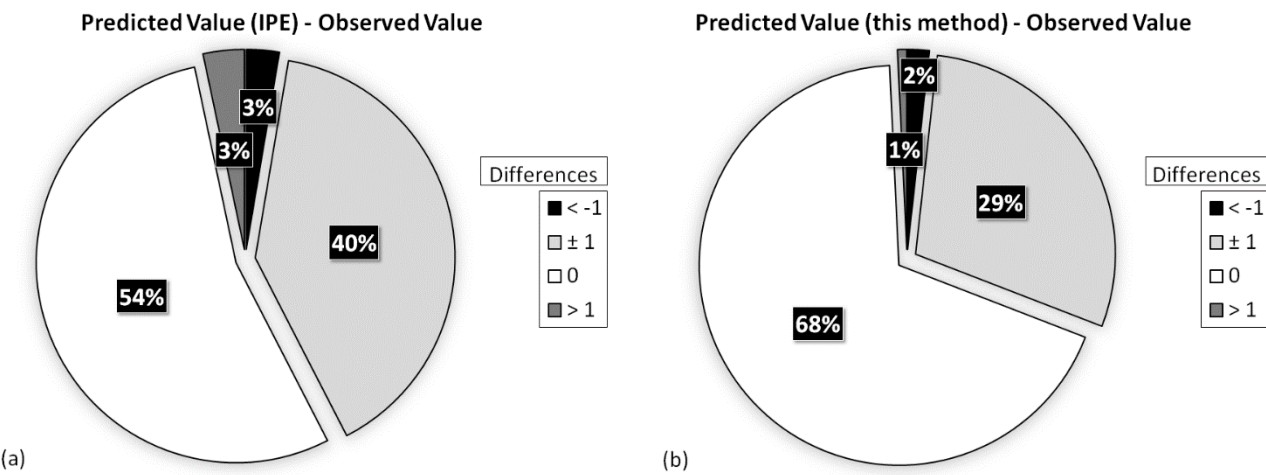

**Figure 7: Differences between the observed intensity values, as reported in DBMI15 (Guidoboni et al., 2007), for the 1980 Irpinia earthquake and the intensity values computed with (a) the IPE alone (Pasolini et al., 2008; Lolli et al., 2019) and (b) the proposed procedure.**

## 5 Case study

How this procedure may serve the purpose of reconstructing the macroseismic fields of past earthquakes, especially those with scattered IDPs, is shown by means of the case study of an earthquake occurred on 13 June 1542 in the Mugello area (Northern Apennines), with Mw 6 and epicentral intensity equal to 9 according to CPTI15. In DBMI15 there are 45 IDPs with maximum intensity equal to 9, as assessed by Guidoboni et al. (2007). As reported in Figure 8, the effects of this earthquake are primarily known in the epicentral area with 31 localities with intensity greater than or equal to 8, whereas the macroseismic information at the localities far from the epicenter is extremely scattered.

With the aim of integrating the intensity distribution of this earthquake, 968 localities of DBMI15 within a radius of 20 km from each of the 45 IDPs were considered. Figure 9 shows the  modal values of the probability distribution $p_I(I_s/I_v)$ computed at each of the 968 localities assuming as prior distribution the probability derived through the IPE (Pasolini et al., 2008; Lolli et al., 2019) and using the intensities observed at all the localities within 20 km (analysis $c$). Such values are compared with the intensities (expressed as modal values) predicted by the IPE alone. Figure 9 shows that the intensity values estimated by the two approaches are quite different, particularly in the epicentral area. For example, focusing on the area where the IPE alone predicts intensity 7, the intensities computed by the proposed procedure are equal to 6, 7 and 8. On the contrary, moving away from the epicentral area, the two approaches provide similar results for intensity 5.

Figure 10 displays the geographical distribution of the predicted intensities at the 968 localities represented as the probability to be greater than or equal to intensity 6 and 8, computed through i) the IPE alone, and ii) the proposed procedure. As shown


in Fig. 10a and 10b, the probability of intensity greater or equal to 6 is more than 90% for 278 localities using the IPE alone, while the same probability extends to 488 localities using the second procedure. The differences between the two approaches

become more evident in case of localities where the probability of intensity greater than or equal to 8 is higher than 50%, that is 96 localities using the IPE alone (Fig. 10c) and 212 using our procedure (Fig. 10d).

The results shown in Fig. 9 and Fig. 10 consent to appreciate the impact of the proposed methodology in reproducing the pattern of observed intensities with respect to the simple isotropic decay of intensity with distance predicted by IPEs.

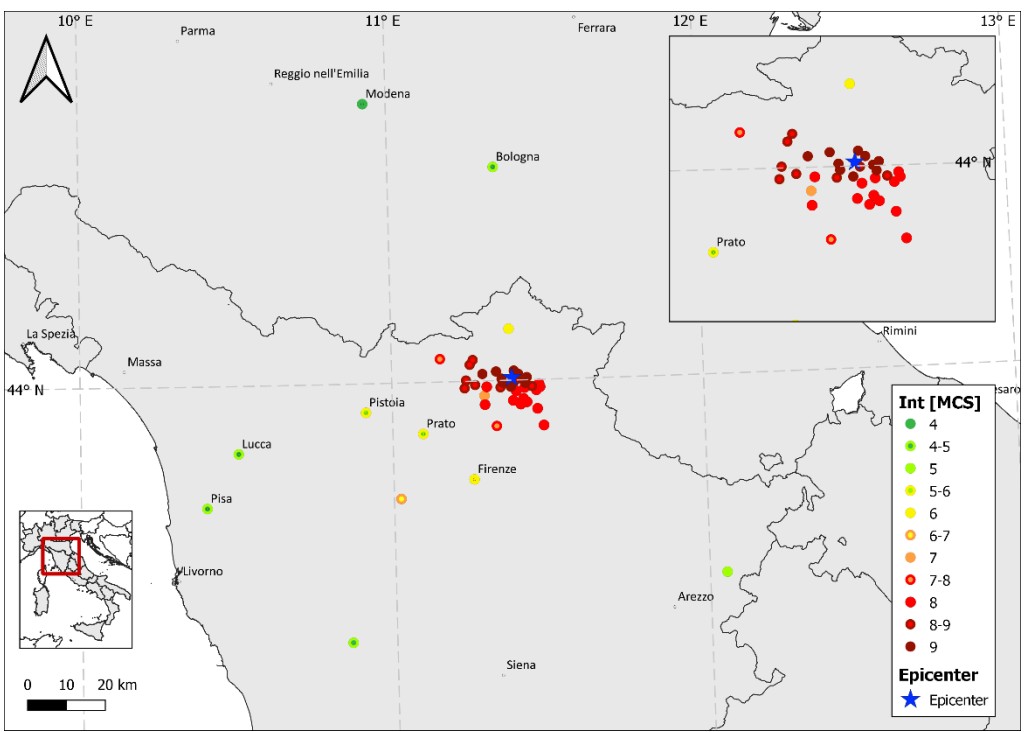

**Figure 8: Intensity distribution of the 1542 Mugello earthquake assessed by Guidoboni et al. (2007) and reported in DBMI15.**

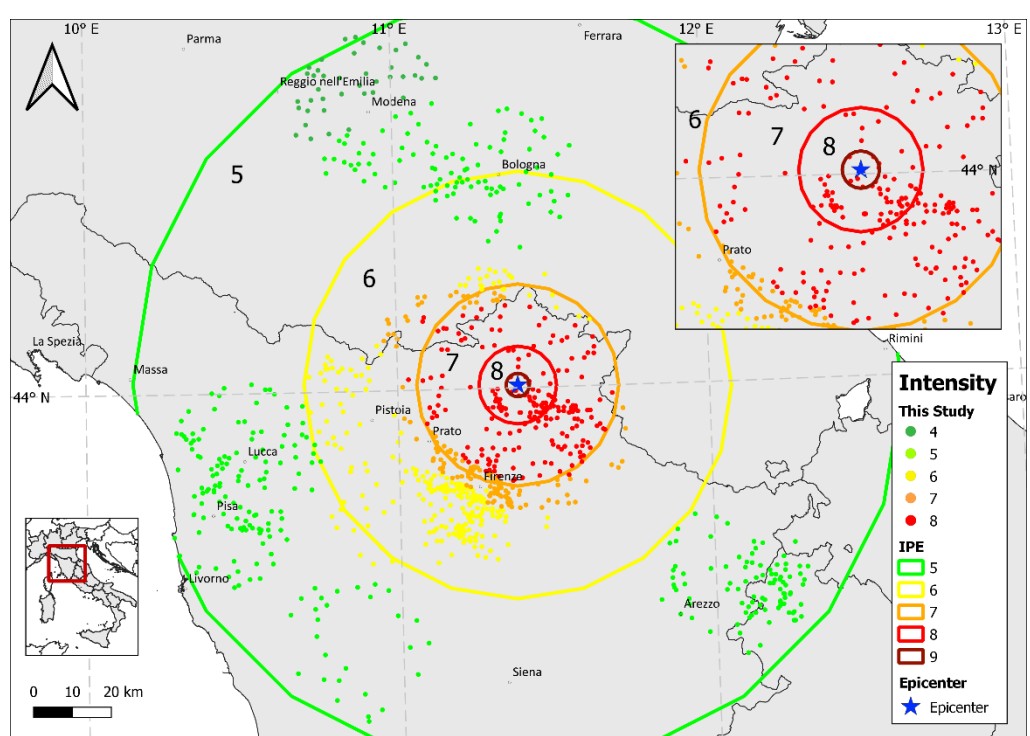

**Figure 9: Modal values of the probability distribution $p_l(I_s|I_v)$ computed at the 968 considered localities (small dots) for the 1542 Mugello earthquake; colored circles bound areas of different intensity values predicted by the IPE.**
**Figure 10: Intensity distribution of the 1542 Mugello earthquake at the 968 considered localities represented as the probability to be greater than or equal to intensity 6 with (a) the IPE alone and (b) this procedure, and to intensity 8 with (c) the IPE alone and (d) this procedure.**

**5 Conclusions**

The procedure proposed in this article estimates the probability distribution for a given intensity value at the considered site through a Bayesian approach. The procedure takes into account (i) region-dependent empirical relations to model macroseismic intensity attenuation with source distance (i.e., IPEs), (ii) probability distributions resulting from the in-depth analysis of the spatial variability of intensity data collected in the Italian Macroseismic Database DBMI15 (section 3.2), and (iii) the discrete

and ordinal nature of macroseismic intensity and its uncertainties. This procedure allows improving the macroseismic intensity distributions of historical earthquakes constraining the intensity values calculated at a site through an IPE with intensities

observed at neighboring localities for the same earthquake.

The results obtained in the application part (see section 4.2) emphasize that this method well reproduces the observed values for intensity greater than or equal to 6 and equal to 3. On the other hand, the outcomes for intensity 4 and 5 show respectively an overestimation and underestimation that could be linked to both: (i) the incompleteness of the analyzed dataset due to the input threshold of DBMI15 (intensity $\geq$ 5) and (ii) the incompleteness of historical documentation for lower intensity degrees.

These outcomes demonstrate that the intensities predicted with the proposed procedure match the observed values better than using the IPE alone.

This procedure is thought to integrate incomplete and scattered intensity distributions while avoiding the isotropic decay of intensity with distance resulting from existing IPEs. Through a more realistic modelling of the pattern of predicted intensities, this procedure takes into account the spatial distribution and variability of observed intensity data to constrain the results. Not

unexpectedly, the obtained results are dependent on the spatial distribution of the data observed for the selected earthquake and on the number of intensity values available in nearby localities.

The proposed procedure aims at the integration and enrichment of both the intensity distributions of individual earthquakes and the seismic history of single localities. Together with suggestions to further document the spatial distribution and severity of effects in the framework of historical seismological investigation, the outcomes provided by this procedure can be used for

local seismic hazard assessment, as well as planning activities aimed at risk mitigation.

**Data availability**

DBMI15 is available at https://doi.org/10.13127/DBMI/DBMI15.2

CPTI15 is available at https://doi.org/10.13127/CPTI/CPTI15.2

**Author contribution**

AA edited most parts of the paper, performed the statistical analyses and tested the results. AR, VD and DA contributed to the manuscript and supervised the research.

**Competing interests**

The authors declare that there is no conflict of interests.



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
