# Peer review of "Integrating macroseismic intensity distributions by a probabilistic approach: an application in Italy"

_Natural Hazards and Earth System Sciences, 2021_

## Referee Comment (RC1)

Comments to the paper "Integrating macroseismic intensity distributions by a probabilistic approach: an application in Italy" by Antonucci et al.

The paper is sound and well written. I suggest it is published after the authors have answered the following questions:

Table 1, I'm very surprised that the q(Delta I)all relative frequencies are perfectly symmetric (to the fifth figure for -1 and +1) while for example the q(Delta I)near ones are not.

The caption of Fig. 1 swaps the two panels: actually left panel refers to localities within 20 km and the right one to localities within 10 km.

Line 279-282. "Figure 9 shows the modal values of the probability distribution pl(Is|Iv) computed 280 at each of the 968 localities assuming as prior distribution the probability derived through the IPE (Pasolini et al., 2008; Lolli et al., 2019) and using the intensities observed at all the localities within 20 km (analysis c). Such values are compared with the intensities (expressed as modal values) predicted by the IPE alone."
I see only one map (maybe with intensities computed using the procedure proposed in the paper) and not a second one (maybe with intensities predicted by the IPE alone). Hence, I cannot make the visual comparison the authors suggest. Please add the second map or (if I misunderstood the argument) please better explain the point. I also do not understand the need of the inset with the central area zoomed by a factor of 1.5 or less.

Line 390. Please provide where (e.g. a web site) the paper by Lolli et al., and more in general the "Il modello di pericolosità sismica MPS19 , rapporto finale" edited by Meletti and Marzocchi, can be found. In my knowledge the latter is not a public domain document.

---

## Referee Comment (RC2)

**Review of the paper titled "Integrating macroseismic intensity distributions by a probabilistic approach: an application in Italy" by Andrea Antonucci , Andrea Rovida , Vera D'Amico and Dario Albarello.**

**General comments**

The paper introduces an improved probabilistic procedure for the estimation of intensities at a site taking into account the availability of intensities at neigboring localities. A large amount of intensity data in Italy is used. The methodology is applied to a historical earthquake with promising results, considering the uncertainties for an earthquake of the 16th century.

The authors have deep knowledge of the procedure, developed in their previous papers, and of the nature of data used.

It is therefore recommended that the paper is published with minor revisions.

Specific comments:
- Do the authors consider all IDPs used in the study have the same quality?
- I would not add a comma after e.g. or after i.e.
- Comment on Figures 8 and 10d: Observed Intensities seem to cover a more limited area than the synthetic with probability >90%. Would this imply for a larger earthquake magnitude? Please comment.
- L62: spatially close to the site of interest: assuming similar local soil conditions at these localities?
- L137-138: Comment: at 20 km distance, the possibility described in lines 128-129 is very low.
- L149: How is intensity 1-2 defined?
- L213: Which time period cover the seismic histories of the 28 localities?
- L213: More info is necessary on the decision for selection of the 28 localities.

Technical corrections:

L16: replace "data" with "values"

L26: add a comma after coordinates

L27: replace "sources" with "source"

L46: Postpischl 1980 not in reference list

L125: "at a pair" *or* "at pairs"

L162: replace "in the following" with "as below"

L414-416: Postpischl et al 1985 not in text

L429-431: Rovida et al 2021 not in text

---

## Author Comment (AC1)

Comments to the paper "Integrating macroseismic intensity distributions by a probabilistic approach: an application in Italy" by Antonucci et al.

The paper is sound and well written. I suggest it is published after the authors have answered the following questions:

Table 1, I'm very surprised that the q(Delta I)all relative frequencies are perfectly symmetric (to the fifth figure for -1 and +1) while for example the q(Delta I)near ones are not.
This is due to the fact that for $q(\Delta I)_{all}$ all the localities within a radius of 20 km from the site are considered, whereas for $q(\Delta I)_{near}$ only the nearest one is taken into account. In case of $q(\Delta I)_{near}$ this means that, for example, if locality B is the nearest one to locality A, locality A might not be the nearest one to B (e.g. the nearest locality is C) and, thus, the $\Delta I$ value between A and B (i.e. $I_B - I_A$) is computed only once. In case of $q(\Delta I)_{all}$, instead, locality B is considered among the localities within 20 km from locality A and vice versa, thus the $\Delta I$ value between A and B is computed twice (i.e. $I_B - I_A$ and $I_A - I_B$ ) and the relative frequencies result symmetric.
In other words, for $q(\Delta I)_{near}$ we can have $I_B - I_A$ and $I_C - I_B$ (not $I_A - I_B$), whereas for $q(\Delta I)_{all,}$ we always have $I_B - I_A$ and $I_A - I_B$.

The caption of Fig. 1 swaps the two panels: actually left panel refers to localities within 20 km and the right one to localities within 10 km.
Thank you, it was a mistake and we will correct it in the revised manuscript.

Line 279-282. "Figure 9 shows the modal values of the probability distribution pl(Is|Iv) computed 280 at each of the 968 localities assuming as prior distribution the probability derived through the IPE (Pasolini et al., 2008; Lolli et al., 2019) and using the intensities observed at all the localities within 20 km (analysis c). Such values are compared with the intensities (expressed as modal values) predicted by the IPE alone."
I see only one map (maybe with intensities computed using the procedure proposed in the paper) and not a second one (maybe with intensities predicted by the IPE alone). Hence, I cannot make the visual comparison the authors suggest. Please add the second map or (if I misunderstood the argument) please better explain the point. I also do not understand the need of the inset with the central area zoomed by a factor of 1.5 or less.
The map in Figure 9 shows both the modal values of the probability distributions computed at the 968 considered localities with our procedure (small dots) and the modal values predicted by the IPE alone. The latter are represented with colored circles that bound areas of different intensity values. In the revised manuscript we will better explain this point. Moreover, we will also remove the inset on the map.

Line 390. Please provide where (e.g. a web site) the paper by Lolli et al., and more in general the "Il modello di pericolosità sismica MPS19 , rapporto finale" edited by Meletti and Marzocchi, can be found. In my knowledge the latter is not a public domain document.
Meletti and Marzocchi (2019) is an internal project report. MPS19 is described in the Meletti et al. (2021; https://doi.org/10.4401/ag-8579) and references therein.

---

## Author Comment (AC2)

**Review of the paper titled "Integrating macroseismic intensity distributions by a probabilistic approach: an application in Italy" by Andrea Antonucci , Andrea Rovida , Vera D'Amico and Dario Albarello.**

**General comments**
The paper introduces an improved probabilistic procedure for the estimation of intensities at a site taking into account the availability of intensities at neigboring localities. A large amount of intensity data in Italy is used. The methodology is applied to a historical earthquake with promising results, considering the uncertainties for an earthquake of the 16th century.
The authors have deep knowledge of the procedure, developed in their previous papers, and of the nature of data used.
It is therefore recommended that the paper is published with minor revisions.

Specific comments:
- Do the authors consider all IDPs used in the study have the same quality?
- The Italian Macroseismic Database – DBMI15 does not provide any quality assessment for the single IDPs. Anyway, some exclusion criteria have been adopted (section 3.2): in particular, we discarded data relative to non-numerical macroseismic observations and unidentified localities/large areas. Moreover, volcanic earthquakes have not been considered in the analysis.
- I would not add a comma after e.g. or after i.e.
- We followed the most recent articles published in NHESS, where there are commas after e.g. and i.e., but we leave the decision to the editorial office.
- Comment on Figures 8 and 10d: Observed Intensities seem to cover a more limited area than the synthetic with probability >90%. Would this imply for a larger earthquake magnitude? Please comment.
- The earthquake magnitude computed considering the synthetic intensities for the case study in Figure 10d could be different from the one calculated from observed ones. However, synthetic intensities cannot be used for estimating earthquake magnitude because they are computed starting from an IPE (and then combined with intensity data at nearby localities), which uses the magnitude of the selected earthquake as independent variable.
- L62: spatially close to the site of interest: assuming similar local soil conditions at these localities?
- We do not make specific assumptions on local soil conditions.
- L137-138: Comment: at 20 km distance, the possibility described in lines 128-129 is very low.
- The search radius was selected by balancing the needs for maximizing the number of intensity data within the radius (related to the average density of settlements in Italy) and minimizing the possible geological heterogeneities present in the same area. As represented in Figure 1, the best balance among these conditions appeared to be 20 km. This point will be better explained in the revised manuscript.
- L149: How is intensity 1-2 defined?
- In DBMI15 there are very few IDPs with intensity equal to 1-2 derived from macroseismic bulletins resulting from macroseismic questionnaires. No explanation is given in the original data. In our dataset these values are very few (i.e. 17 in this case) and they do not affect the results.
- L213: Which time period cover the seismic histories of the 28 localities?
- L213: More info is necessary on the decision for selection of the 28 localities.

- The considered seismic histories cover the same time-period of DBMI15, i.e. 1000-2017, but their length varies for each locality. In the revised version of the manuscript, we will add a new table with the details of the seismic histories of the 28 selected localities, such as the number of IDPs, the time coverage, the maximum intensity and we will also show their location in Figure 1.

Technical corrections:
L16: replace "data" with "values"
L26: add a comma after coordinates
L27: replace "sources" with "source"
L46: Postpischl 1980 not in reference list
L125: "at a pair" *or* "at pairs"
L162: replace "in the following" with "as below"
L414-416: Postpischl et al 1985 not in text
L429-431: Rovida et al 2021 not in text
We thank the reviewer for all the technical corrections, which we will take into account in the revised manuscript.

---

## Author Response (AR1)

**Authors' Response**

We accepted all the comments and suggestions provided by the reviewers. We modified the text as follow:

- Section 3.2: we added one sentence to better explain the distance considered for characterizing the probability $q(I_v|I_s)$, as requested by RC2;
- We modified Figure 1 according to RC1 and we also added the 28 test localities used in section 4 according to RC2;
- Section 4.2: we added a new table (Table 2) with the details of the seismic histories of the 28 test localities following RC2;
- We modified Figure 8 and Figure 9 according to RC1. We also added one sentence in the text to better explain the Figure 9.

In addition, we modified Figure 6 and the related text because we realised that there was a little mistake in the values. This change does not influence the results of the analysis.